# A Three-Dimensional Nickel(II) Framework from a Semi-Flexible Bipyrimidyl Ligand Showing Weak Ferromagnetic Behavior

**DOI:** 10.3390/polym11010119

**Published:** 2019-01-11

**Authors:** Shin-Shan Dong, Chen-I Yang

**Affiliations:** Department of Chemistry, Tunghai University, Taichung 407, Taiwan; nmr9902030@gmail.com

**Keywords:** semi-flexible bipyrimidyl ligand, coordination polymer, magnetic properties, spin-canting, magnetic ordering, weak ferromagnetism

## Abstract

A semi-flexible bipyrimidyl ligand, 5,5′-bipyrimidin (bpym), was used for the self-assembly of a transition metal coordination polymer, resulting in the formation of a nickel(II) compound, [Ni(Br)_2_(bpym)_2_]*_n_* (**1**) with a three-dimemsional (3D) structure. Single-crystal X-ray analysis showed that compound **1** crystallizes in the monoclinic space group *C*2/c and the structure represents a 3D (4,4)-connected **bbe** topological framework constructed of nickel(II) ions, twisted *cis*-*μ*-bpym and planar *trans*-*μ*-bpym groups. Magnetic characterization revealed that **1** shows antiferromagnetic coupling between the pyrimidyl-bridged Ni(II) ions along with weak ferromagnetism due to spin canting with a magnetic ordering below *T*_c_ = 3.4 K.

## 1. Introduction

The design and synthesis of novel multi-dimensional metal-organic frameworks (MOFs) and/or coordination polymers (CPs), which could be function as molecule-based magnets, has attracted considerable attention in recent years due to their fascinating and diverse structures and topologies that could play a role in a variety of magnetic interactions and resulting bulk magnetic behavior [1,2,3,4,5,6,7,8]. The majority of synthetic strategies for preparing such magnetic CPs involve the use of paramagnetic ions as building blocks and bonding them in close proximity to one another via short bridging ligands, a configuration that would allow for significant magnetic exchange. The nature of the paramagnetic ions means the bulk magnetic behavior of the magnetic CPs also depend on several additional factors, including the magnitude and nature of the magnetic interaction transmitted by the bridging ligands, the structure of the extended network, and cooperative interactions between spin carriers. Therefore, enormous efforts have been focused on searching in exploring appropriate bridging ligands for that can be used in the construction of magnetic CPs with various extended networks.

Regarding bridging ligands for magnetic CPs, six-membered heterocyclic diazines, such as pyridazine, pyrazine, pyrimidine and derivatives thereof have been subjects of considerable interest because they strongly donate *σ*-electrons to metal centers and the fact that they can exist in various coordination modes, which allow the formation of numerous CPs with diverse extended structures and topologies that exhibit different bulk magnetic properties such as spin-canted antiferromagnetism, ferromagnetism, metamagnetism, ferrimagnetism [9,10,11,12,13,14,15,16]. Furthermore, a 2,2’-bipyrimidine unit, when fused with two 2-Br-substituted pyrimidines, has been used as a bridging ligand for producing numerous magnetic CPs with various structural topologies and magnetic properties, in which the pyrimidyl groups not only connect multiple metal centers to one another, leading to extended architectures but also mediate significant magnetic interactions, thus conferring unusual bulk magnetic behavior, such as spin canting and metamagnetism [17,18,19]. However, CPs from 2,2’-bipyrimidine usually have a low-dimensional structure (0D, 1D and 2D) because the ligand lacks flexibility by strong chelating effects.

Following this concept, we selected 5,5’-bipyrimidine (bpym), a semi-flexible bipyrimidyl ligand, from a combination of two 5-Br-substituted pyrimidine derivatives, as a bridging ligand for preparing a series of new magnetic CPs. We hypothesized that this ligand would confer several characteristics when it was coordinated by metal ions; (i) the ligand has four potentially coordinated *N*-donor sites, it would be expected to exhibit various bridging modes and would connect metal centers to produce extended structures; (ii) the ligand has multiple structural geometries. The dihedral angles of two out of the plane aromatic pyrimidyl rings provide sufficient adaptability and flexibility to satisfy the requirements needed for the assembly of coordination structures with diverse frameworks; (iii) the presence of multiple nitrogen atoms may assist in the construction of supramolecular structures through aromatic π–π interactions and hydrogen bonding; (iv) bridging through a pyrimidyl group (–N–C–N–) may, not only result in a shorter the distance between the adjacent metal centers but would also mediate significant magnetic couplings.

Although five examples of CPs with bpym ligand have been reported [20,21,22,23], examples of multi-dimensional magnetic CPs based on the bpym ligand are still rare. Therefore, in a previous study, we selected bpym as a candidate for combining with azido anions to synthesise two new series of metal-azide-bpym magnetic CPs [24,25]. In this study, we report on the use of bpym to exclusively react with the NiBr_2_ without any co-bridging groups, resulting in the formation of a new 3D Ni(II) coordination polymer, [Ni(Br)_2_(bpym)_2_]*_n_* (**1**). Compound **1** features a 3D framework comprised of 1D nickel–pyrimidyl chains that are aligned parallel in the [001] direction and represents a rare **bbe** topology. Magnetic characterizations revealed that compound **1** shows weak ferromagnetism due to spin canting and magnetic ordering with a critical temperature (*T*_C_) of 3.4 K.

## 2. Experimental

### 2.1. Materials and Methods 

All reagents and solvents were used as received without further purification. All of the reactions were carried out under aerobic conditions. The 5,5′-bipyrimidine was prepared following a previously reported procedure [26].

### 2.2. Synthesis of [Ni(Br)_2_(bpym)_2_]_n_ (**1**)

A light green methanol solution (5 mL) of NiBr_2_·H_2_O (30.1 mg, 0.13 mmol) was carefully layered on top of a colorless aqua solution (5 mL) of bpym (10 mg, 0.06 mmol), and NaBr (26.1 mg, 0.25 mmol), and the resulting solution was allowed to stand at room temperature. After two weeks, block light blue crystals of **1** were formed. The formed crystals were washed with water, collected by suction filtration and dried in air. Yield: 80% (based on bpym). The powder X-ray diffraction pattern measured from the bulk sample matched well with the simulation pattern from the single-crystal data (vide infra). Elemental analysis calcd (%) for C_32_H_24_Co_2_N_1__6_ (**1**): C, 20.95; H, 2.26; N, 35.93. Found: C, 20.70; H, 2.16; N, 36.02. IR data (KBr disk, cm^−1^): 3414(w), 3039(w), 1590(s), 1580(s), 1557(vs), 1449(m), 1406(vs), 1354(vs), 1340(s), 1199(m), 1188(m), 1153(w), 1144(w), 1052(w), 1010(s), 985(w), 971(w), 929(w), 903(s), 719(vs), 687(s), 667(vs), 650(s), 634(s).

### 2.3. X-ray Crystallography

Diffraction intensity data of compound **1** were collected at 150 K on a Brucker APEXII CCD diffractometer (Bruker, Bruker, Karlsruhe, Germany) with graphite-monochromated Mo Kα radiation (*λ* = 0.7107 Å). The program SADABS (Bruker, 2016) was used for absorption corrections [27]. The structure was solved by direct methods and refined using the full-matrix least-squares method against F^2^, using the SHELXTL-2014 program [28]. The refinements of all non-hydrogen atoms were treated anisotropically with thermal parameters, whereas the hydrogen atoms on their respective carbon atoms were placed in ideal, calculated positions, using a riding model with isotropic thermal parameters. Experimental details for X-ray crystallographic data and the refinements of compound **1** are summarized in Table 1 and the selected bond distances and angles are listed in Table 2.

### 2.4. Physical Measurements

The temperature dependence direct current (dc) and alternating current (ac) magnetic susceptibility measurements were collected on powdered samples of **1** on a Quantum Design MPMS-7 SQUID and a Quantum Design PPMS (Physical Property Measurement System) magnetometer (Quantum Design, San Diego, CA, USA) with a 7.0 T and 9.0 T equipped magnets, respectively, operated in the temperature range of 2.0–300 K. To prevent torqueing, the powdered samples were restrained in eicosane before measurement. The Pascal’s constants were estimated for diamagnetic corrections [29], which were subtracted from the experimental susceptibility to achieve the molar paramagnetic susceptibility of **1**. Elemental analyses (carbon, hydrogen, nitrogen) of compound **1** were performed using an Elemental vario EL III analyzer (PerkinElmer, Taipei, Taiwan). Thermogravimetric (TG) analysis of compound **1** was measured on a Seiko Instrumental, Inc. (Chiba shi, Japan), EXSTAR 6200 TG/DTA analyzer, preforming in a heating rate of 5 °C/min under an atmosphere of nitrogen. Powder X-ray diffraction (PXRD) measurement of **1** was carried out on a Siemens D-5000 diffractometer running in a step mode with a step size of 0.02° in θ and a fixed time of 10 s at 40 kV, 30 mA for Cu-Kα (*λ* = 1.5406 Å). Fourier transform infrared (FTIR) spectra were measured by a Perkin-Elmer Spectrum RX1 FTIR spectrometer with KBr pellets (PerkinElmer, Taipei, Taiwan).

## 3. Results and Discussion

### 3.1. Syntheses and Characterization of Compound **1**

Compound **1** was prepared by reacting nickel(II) bromide, sodium bromide and the bpym ligand in a MeOH/water solution at room temperature. In the initial procedure, the reaction was carried out by mixing the NiBr_2_ and bpym in MeOH without NaBr, resulting in the formation of only a few blue crystals in ~10% yield, which were identified by comparing the infrared (IR) spectrum with compound **1** and by unit cell determination. In subsequent experiments, additional NaBr was added to the reaction mixtures of NiBr_2_ and bpym in MeOH to enhance the molar ratio of Br^−^/Ni(II) ions, which resulted a much higher yield of crystalline **1** (~80%). This result indicates that the presence of an excess of Br^−^ anions in the reaction leads to higher yields of the **1**.

The phase purity for the bulk material **1** was independently established by elemental analysis, and PXRD (Appendix A). The thermal stability of compound **1** is also examined by TG analysis (Appendix A), in which a large weight loss was observed in the TG curve at about 230 °C, indicating the decomposition of the bpym ligands and the frameworks. Based single-crystal X-ray diffraction and the results of elemental analysis, the formula for the compound was determined to be [Ni(Br)_2_(bpym)_2_]*_n_*. Compound **1** exhibited several strong IR bands around *ν* = 1590−1557 cm^−1^, which were assigned to *ν*(C=Ν) and *ν*(C=C) vibrations characteristic of the bpym ligand.

### 3.2. Description of Structure

#### 3.2.1. Crystal Structures of Compound **1**

X-ray analysis results revealed that compound **1** crystallizes in the monoclinic space group *C*2/c and the asymmetric unit contains two half crystallographically independent Ni(II) ions (Ni1 and Ni2), two bpym ligands and two half and one crystallographically independent coordinated Br^−^ anions (Br1, Br2, and Br3). As depicted in Figure 1, the two crystallographically distinct Ni(II) ions both show a six coordination of the NiN_4_Br_2_ coordination environment in an elongated distorted octahedral geometry, where four nitrogen atoms (N6, N7 and their symmetrical equivalents for Ni1; N3, N5 and their symmetrical equivalents for Ni2) from the bpym ligands and two Br^−^ anions (Br1, Br2 for Ni1; Br3 and its symmetrical equivalent for Ni2) occupy equatorial and axial positions, respectively. The Ni centers in **1** have Ni−N bond distances of 2.100(2) to 2.139(2) Å, Ni−Br bond distances of 2.572(3) to 2.5884(6) Å, *cis* N−Ni−N angles of 88.61(9) to 95.13(13)°, *cis* Br−Ni−N angles of 87.70(6)−94.51(6)°, *trans* N−Ni−N angles of 177.25(8)−179.18(12)° and *trans* Br−Ni−Br angles of 173.40(2)−180.000(1)°, which are consistent with the values for the reported Ni(II) compounds [30,31,32]. Compound **1** contains two types of bridging bpym ligands, one of which adopts a *syn*-*μ*-bridging mode with two pyrimidyl rings twisted in 40.20(4)° that connect the Ni centers through its two N atoms (N7 and N8) in one pyrimidyl ring (Scheme 1, mode A). The other one exhibits an *anti*-*μ*-bridging mode with two coplanar pyrimidyl rings linking the Ni centers by two N atoms from two pyrimidyl rings (Scheme 1, mode B). The Ni1 and Ni2 metal centers are bridged by the twisted *syn*-*μ*-bpym ligands leading to the formation of a screw-type of 1D Ni_2_(*μ*-pym)_2_-based coordination chain with adjacent Ni1···Ni2 distance of 5.925(1) Å. In addition, the NiN_4_ equatorial planes of the Ni1 and Ni2 polyhedrons are perpendicular to each other. Each Ni_2_(*μ*-pym)_2_-based chain is inter-linked to three neighboring chains bridged by the planar *anti*-*μ*-bpym ligands, resulting in the formation of a (3D) coordination framework (Figure 2 and Figure 3), with interchain Ni···Ni distances of Ni1···Ni1A and Ni2···Ni2A are 9.7429(7) Å and 9.8148(7) Å. The presence of both *syn*-*μ*-bpym and *trans*-*μ*-bpym bridging ligands in **1** makes it different from other known coordination compounds that contain bpym ligands [20,21,22,23].

To obtain a additional insights into the reason for the 3D framework structure of **1**, a topological analysis is performed. As described above, each Ni1 atom was bonded by two twisted *syn*-*μ*-bpym and two planar *anti*-*μ*-bpym ligands which are in *trans*-positions relative to the equatorial planes, which can be considered as a pyramidal 4-connected node when the terminal coordinated Br^−^ anions are omitted (Figure 3a). Similarly, each Ni2 center is linked by two twisted *syn*-*μ*-bpym and two planar *anti*-*μ*-bpym ligands in *cis*-positions relative to the equatorial planes and, thus, can be viewed as a planar 4-connected node by omitting the terminal coordinated Br^−^ anions (Figure 3b). On the other hand, each *syn*-*μ*-bpym and *anti*-*μ*-bpym ligand can be regarded as 2-connectors. The entire framework of **1** was accordingly simplified as illustrated in Figure 4. An analysis of **1** using the TOPOS program [33], indicates that the framework of **1** can be rationalized as a rare binodal (4,4)-connected **bbe** topology with the Schläfli symbol (4^6^.4^8^)( 4^3^.6^3^) as illustrated in Figure 4. To the best of our knowledge, compound **1** appears to be the first example of a metal coordination polymer with a **bbe** topology.

### 3.3. Magnetic Properties

The temperature dependences for magnetic susceptibility were measured on powdered samples of compound **1** in the temperature range from 2 to 300 K under an applied field of 1000 Oe of applied field. The temperature dependence of *χ*_M_ and *χ*_M_*T* plots of compound **1** are shown in Appendix A and Figure 5, respectively. As the temperature decreases from 300 K, the *χ*_M_ value increases smoothly, reaching a rounded maximum of 0.044 emu mol^−1^ at about 9.0 K, and then decreases slightly reaching a value of 0.041 cm^3^ mol^−1^ at 4.0 K. Upon further cooling, the *χ*_M_ value increases rapidly to a sharp maximum of 0.053 cm^3^ mol^−1^ at 2.5 K, and then decreases to 0.052 cm^3^ mol^−1^ at 2.0 K. The temperature dependence of 1/*χ*_M_ at temperatures above 50 K can be fitted by the Curie–Weiss law with a Curie constant *C* = 1.26 cm^3^ mol^−1^ K and a Weiss constant *θ* = −9.63 K (Appendix A). The negative Weiss constant suggests the presence of overall antiferromagnetic interactions between the adjacent Ni(II) ions. At 300 K, the *χ*_M_*T* value per Ni(II) of **1** is 1.21 cm^3^ mol^−1^ K, which is larger than the spin-only value of 1.00 cm^3^ mol^−1^ K for a magnetically isolated octahedral Ni(II) ion (S = 1), with g = 2.00. Upon cooling, the *χ*_M_*T* decreases monotonically, eventually reaching a minimum value of 0.140 cm^3^ mol^−1^ K at 3.5 K, which is indicative of the existence of antiferromagnetic coupling between the magnetic centers. After a very small y increase to a maximum value of 0.147 cm^3^ mol^−1^ K at 3.0 K, the *χ*_M_*T* value then decreases upon further cooling to 2.0 K. The slight increase in *χ*_M_*T* below 3.5 K suggests that a mechanism involving ferromagnetic correlations is operative within compound **1** and the final decrease may be attributed to antiferromagnetic interactions between the chain and/or saturation effects. This ferromagnetic correlation can be attributed to spin-canted antiferromagntism. For a spin canting material, the antiferromagnetically coupled local spins are not perfectly antiparallel but, rather, are canted with respect to each other, which results in uncompensated residual spins.

Spin canting is generally attributed to the existence of (1) antisymmetric superexchange interactions (Dzyaloshinsky−Moriya interaction) and/or (2) the appropriate single-ion magnetic anisotropy of metal ions. These two factors could induce a different preferential direction in the magnetic moments for metal ions that are located in different sublattices [34,35]. The antisymmetric exchange coupling would be predicted to disappear in the presence of an inversion center between adjacent spin centers [35]. Taking the structural features of **1** into account for magnetic properties, the existence of spin canting is consistent with coordination polyhedral orientations that systematically alternate through the chain, with the absence of an inversion center between the neighboring Ni(II) centers bridged by the *μ*-pym group. Moreover, a weak magnetic anisotropy might result from the distorted octahedral Ni(II) metal center, which could also be a contributor of the spin canting in **1**. Similar spin canting behavior due to antisymmetric magnetic interactions and the weak magnetic anisotropy of metal ions have been observed for other Ni(II) compounds [36,37].

In analyzing the magnetic interaction between the Ni(II) of compound **1** in the high temperature range, magnetic interactions through planar *anti*-*μ*-bpym bridges would be expected to be weak, because of the long distance between the Ni ions (Ni···Ni > 9.7 Å). Thus, compound **1** can be approximately rationalized as a “uniform Ni-pyrimidyl 1D chain system” and the temperature-dependent magnetic susceptibility of **1** can be fitted by the Fisher chain model with *S* = 1 based on the Hamiltonian *H* = −*JS_i_⋅S_i_*_+1_ [38] Equation (1):*χ* = [*Ng*^2^*β*^2^*S*(*S* + 1)/(3*kT*)] [(1 + *u*)/(1 − *u*)](1)
where *N*, *β* and *k* are Avogadro’s number, Bohr’s magneton*,* and Boltzmann’s constant, respectively, and *u* is the well-known Langevin function (Equation (2)): *u* = *L*(*JS*(*S* + 1)/*kT*) = coth(*JS*(*S* + 1)/*kT*) − *kT*/*JS*(*S* + 1)(2)

The best fit above 10 K leads to parameters of *J* = −5.77 cm^−1^, *g* = 2.2 and *R* = 1.7 × 10^−4^, were *R* = ∑[(*χ*_M_*T*)_calcd_ − (*χ*_M_*T*)_obs_]^2^/∑(*χ*_M_*T*)^2^_obs_. The negative value for *J* indicates that antiferromagnetic interactions are dominated through the pyrimidine moiety between the Ni(II) centers, and is located in the range of previously reported values for pyrimidine-bridged Ni(II) compounds [39,40].

In order to further substantiate the weak ferromagnetism associated with the spin canting in **1**, magnetic susceptibilities at different applied fields (10, 50, 100, and 1000 Oe) in the temperature range of 1.8–10 K were collected. As can be seen in Figure 6, *χ*_M_*T* increases abruptly at temperatures below 3.6 K for all fields, and the increases in *χ*_M_*T* values at low-temperatures become less pronounced at higher fields, thus confirming the existence of a weak ferromagnetic state due to spin canting antiferromagnetism. Moreover, to obtain additional evidence for the magnetic ordering of **1**, zero-field-cooled/field-cooled (ZFC/FC) magnetization studies were carried out. As shown in Figure 7, upon cooling, both ZFC and FC magnetizations increase abruptly at temperatures below 3.6 K and a divergence between ZFC/FC below 3.4 K is observed, indicating that **1** undergoes spontaneous magnetization due to the magnetic ordering of the canted spins and the formation of an ordered state below the critical temperature (*T*_c_) of 3.4 K. The results obtained from the ZFC/FC magnetization studies are compatible with the observed ac magnetic susceptibility. The ac magnetic susceptibility measurements were performed under *H*_dc_ = 0 Oe and *H*_ac_ = 3.5 Oe at different frequencies (Figure 8). As can be seen in Figure 8, both in-phase (*χ*_M_′) and out-off phase (*χ*_M_″) signals are frequency-independent and show maxima at *ca.* 3.4 K, thus confirming the occurrence of magnetic ordering by weak ferromagnetism due to spin canting, which is consistent with the ZFC/FC magnetization results. The presence of non-zero *χ*_M_″ signals below *T*_c_ are the result of the formation of an uncompensated moment and, consequently, a coercive magnetic behavior would be expected.

In an attempt to obtain additional evidence for the weak ferromagnetism of spin canting, the isothermal magnetization *M*(*H*) of **1** was examined at 2.0 K up to 50 kOe (Figure 9a). The magnetization increased steadily to 0.38 *Nβ* at 50 kOe without achieving saturation, which is much lower than the expected saturation value of 2.0 *Nβ* anticipated for an isotropic Ni(II) ion (*S* = 1, *g* = 2.0), thus confirming the existence of spin canting antiferromagnetism. In a careful examination of the data in the low-field region, a small hysteresis loop was observed at 2.0 K (Figure 9b), which shows a coercive field (H_c_) of about 200 Oe and a remnant magnetization of 0.0032 *Nβ*. Based on the remnant magnetization, the canting angle was estimated to be approximately 0.18° [41,42].

Solid state X-band electron paramagnetic resonance (EPR) spectra of a powdered sample of compound **1** was collected at 77 K (Appendix A). The spectrum shows a weak and inhomogenously broadened band, where the peak-to-peak Δ*H*_p-p_ is 1945 G, and the center is at 2615 G. The EPR signal obtained represents a typical spectrum for octahedral Ni(II) complexes. This inhomogenous and broadened peak suggests a significant anisotropic *g*-factor and exchange coupling between each of the Ni(II) ions in **1**.

## 4. Conclusions

In summary, we report on the preparation and characterization of a new type of 3D nickel coordination polymer using a semi-flexible organic ligand, 5,5′-bipyrimidin (bpym). This organic ligand has not been used extensively in the synthesis of coordination polymers. In compound **1**, the distorted octahedral coordinated Ni(II) ions are bridged by *syn*-*μ*-bpym ligands to a Ni(II)-pym chain and the chains are further linked to a 3D framework with a **bbe** topology through *anti*-*μ*-bpym bridges. Magnetic investigations show that compound **1** exhibits weak ferromagnetism due to spin canting with a magnetic ordering of *T*_N_ = 3.4 K. The spin canting in **1** is attributed by antisymmetric magnetic coupling between pyrimidyl-bridged Ni(II) centers and by the presence of single-ion anisotropy of Ni(II) ions. The studies demonstrate that the 5,5′-bipyrimidin is particularly promising for use in the preparation of multi-dimensional coordination polymers with versatile structural topologies and special magnetic properties.

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
