# Peer review of "A Three-Dimensional Nickel(II) Framework from a Semi-Flexible Bipyrimidyl Ligand Showing Weak Ferromagnetic Behavior"

_polymers, 2019, doi:10.3390/polym11010119_

Round 1

Reviewer 1 Report

Polymers 406027

Title: “A three-dimensional nickel(II) framework from a semi-flexible bipy  ramidyl ligand showing weak ferromagnetic behavior”

In this paper, the authors show the synthesis and characterization of a new Ni(II) compound, obtaining by reaction of NiBr2, 5,5’-bipyrimidin and NaBr. The compound was characterized by elemental analysis, IR spectrum and X-ray crystal structure. The magnetic study was presented and this study revealed that this compound is antiferromagnetic with weak ferromagnetism.

In my opinion, this paper is not sufficiently quality for its publication in Polymers. Synthesis and characterization of new compounds is recommended to complete this study.

Comment:

1)      Page 1, Line 11

The formula of the compound is [NiBr4(bpym)2]n or is [NiBr2(bpym)2]n?

Revise all manuscript

2)      Page 1, Line 11

Delete: unique

3)      Page 1, Line 13

CHANGE      nickel(ii)          FOR    nickel(II)

Revise all manuscript

4)      Page 1, Line 21

CHANGE      (CPS) which   FOR    (CPS), which

5)      Page 1, Line 31

CHANGE      for and            FOR    in

6)      Page 3, Table 2

Revise the presentation of the Table 2

7)      Page 4, Line 99

Delete magnetic susceptibility

8)      Page 4, Line 115-120

Without the presence of KBr , which is the structure of the compound 1?

The authors claim that is with KBr that them obtaining 3D structure

9)      Page 4, Line 127

CHANGE      chartacatieristics         FOR    characteristics

10)  Page 5, Line 141

CHANGE bpym liands                      FOR    bpym ligands 

11)  Page 5

Revise the presentation of the Figure 1

12)     Page 5, Line 146

Add references the papers with values of distances and angles

13)  Page 6, Line 175

The paragraph: “To the best of our·······with a bbe topology”

The authors in this paragraph comment that the compound 3 is the first example with bbe topology. Which is the compound 3? This compound is not described in the literature.

14)  Page 8, Line 222-223

“Similar magnetic behaviors·····Ni(II) compounds”

Complete this information. The compounds present the same structure that the compound 1? The core of the complexes described in the literature is the same that the compound 1? The distance Ni····Ni is similar?

15)  Page 13, Line 280-282

Revise the affirmation: “The spin canting····of Ni(II) ions”

16)  Revise the presentation of the references:

[3] Chem. Soc. Rev.

[9] Dalton Transiaction or Dalton Transactions

[10] Dalton Transiton or Dalton Transactions

[31] Magn. Magn. Mater.

[32] Dalton Transitions or Dalton Transactions

[33] J. solid state Chem.

Author Response

1.      P1, L10; P2, L64 and L74; P5, L137. The formula of compound 1 has been corrected to [Ni(Br)2(bpym)2]n.

2.      P1, L10. The “unique” has be deleted as referee’s suggestion.

3.      P1, L13. The “nickel(ii)” has been corrected to “nickel(II)” as referee’s suggestion.

4.      P1, L21. The “(CPs) which” has been corrected to “(CPs), which” as referee’s suggestion.

5.      P1, L31. The “for and” has been changed to “in” as referee’s suggestion.

6.      P3, the presentation of the Table 2 has been revised as referee’s suggestion.

7.      P4, L106. The “magnetic susceptibility” has been deleted as referee’s suggestion.

8.      P5, L124-131. The sentences “In the initial procedure … In subsequent experiments, … 3D structure of 1.” have been changed to “In the initial procedure, the reaction was carried out by mixing the NiBr2 and bpym in MeOH without NaBr, resulting in the formation of only a few blue crystals in ~10% yield, which were identified by comparing the IR spectrum with compound 1 and by unit cell determination. In subsequent experiments, additional NaBr was added to the reaction mixtures of NiBr2 and bpym in MeOH to enhance the molar ratio of Br-/Ni(II) ions, which resulted a much higher yield of crystalline 1 (~8o%). This result indicates that the presence of an excess of Br- anions in the reaction leads to higher yields of the 1.” to refer referee’s suggestion.

9.      P5, L138-139. The sentence “which were assigned ton(C=N) and n(C=C)vibration chartacatieristics of the bpym ligand.” has been changed to “were assigned ton(C=N) and n(C=C) vibrations characteristic of the bpym ligand.” to refer referee’s suggestion.

10.  11. P5, L147. The presentation of Figure 1 has been revised to “As depicted in Figure 1, the two crystallographically distinct Ni(II) ions both show a six coordination of the NiN4Br2 coordination environment in an elongated distorted octahedral geometry, where four nitrogen atoms (N3, N5, N6, N7 and their symmetrical equivalents) the bpym ligands and two Br- anions (Br1, Br2 and Br3 and their symmetrical equivalents) occupy equatorial and axial positions, respectively.”.

12.  Three additional references of Ni(II) complexes coordinated by pyrimidine and bromide ligands were cited and added to refs. 30-32 to refer referee’s suggestion.

13.  P6, L193. The “compound 3” has been addressed to “compound 1”.

14.  P9, L244-246. The sentence of “Similar magnetic behaviors have been obtained for other Ni(II) compounds” has been addressed to “Similar spin canting behavior due to antisymmetric magnetic interactions and the weak magnetic anisotropy of metal ions have been observed for other Ni(II) compounds.” to refer referee’s suggestion.

15.  P12, L313-315. The affirmation of “The spin canting … of Ni(II) ions.” gas been changed to “The spin canting in 1 is attributed by antisymmetric magnetic coupling between pyrimidyl-bridged Ni(II) centers and by the presence of single-ion anisotropy of Ni(II) ions.” as referee’s suggestion.

16.  P12, L334, L335, L336, L344; P13, L347, L349, L350, L352, L355, L356, L358, L361, L362, L364, L365, L368, L371, L375, L379, L386, L390, L393, L394, L397, L398, L399; P14, L405, L406, L408, L426, L431, L436, L441, L444, L447, L451; P15, L454, L456, L457. The presentations of the references haves been revised as referee’s suggestion.

Reviewer 2 Report

Thank you for the efforts of authors. They explained the magnetic properties of compound 1 by SQUID, X-ray analysis and Calculations. However I think this paper will be improved by adding a EPR experimental data and discussion below and above 4K if they can be done. And the magnetic behavior will be clearly explained and suggested the change of electronic state of it around 4K by EPR results. 

Author Response

To refer reviewer’s suggestion, the solid state EPR spectrum of compound 1 was measured. However, the spectrum only recorded at 77 K due to limitation of our EPR equipment. The spectrum of compound 1 was added to Figure S5, and a paragraph of discussion for the observed spectra “Solid state X-band EPR spectra of a powdered sample of compound 1 … Ni(II) ions in 1.” was also added to P. 12, L301-305.

Reviewer 3 Report

This manuscript describe a nickel(II) compound, [Ni(Br)4(bpym)2]n with a unique three-dimemsional (3D) structure and shows its interesting magnetic property. The study on the structural characteristics and magnetic behaviour is well described. The paper is considered to be accepted after minor revision.

Some typing errors:

line 158, "---makes if different from other" should be changed to " ---makes it different from other".

line 175, "compound 3 cappears to ---" should be changed to "compound 3 appears to ---".

Author Response

1. P6, L175-176. The sentence “The presence makes if different from other known coordination …” has been addressed to “The presence of both syn-m-bpym and trans-m-bpym bridging ligands in 1 makes it different from other known coordination compounds that contain bpym ligandsto refer referee’s suggestion.

2. P6, L193-194. The sentence “To the best of our knowledge, compound 3 cappears to be the first example of a metal coordination polymer with a bbe topology.” has been addressed to “To the best of our knowledge, compound 1 appears to be the first example of a metal coordination polymer with a bbe topology.” to refer referee’s suggestion.

Reviewer 4 Report

The manuscript is clearly written and contains quite interesting information about the new magnetic material based on a complex of Ni(II) with a bipyrimidine ligand. Although, in general, the conclusions are well substantiated, the origin of weak ferromagnetism is explained very confusingly. It is clear that the neighboring nickel complexes in the chain are not equivalent: this is evident even from the X-ray diffraction data (Table 2).

The next sentence (lines 216 – 221) should be divided into at least 2 sentences.

 “Taking the  structural features of 1 into account for magnetic properties, although compound 1 crystallizes in a central symmetric crystal system, the existence of spin canting is consistent with the coordination polyhedral orientations are systematically alternation through the chain, with the absence of inversion center between the neighboring Ni(II) centers bridged by the m-pym group.”

The wrong estimation was performed for chi*T value at high temperature. 

Thus, the sentence in lines 200-202 should be replaced by

“At 300 K, the cMT value per Ni(II) of 1 is 1.21 cm3 mol-1 K, which is larger than the spin-only value of 1.0  cm3 mol-1K for magnetically isolated octahedral Ni(II) ion (S = 1), with g = 2.0.

Small corrections:

The numbering in the Table 2 should coincide with the numbering in Fig. 1.

Line 58, “in a previously study” change to “in a previous study”/

Line 175, “3 appears”

Line 194, “smooth” -> “smoothly”

Line 281, “pyrimidy” -> “pyrimidyl”

Line 287, ”.

Author Response

1.      P8, L233-P9, L242. The sentences “In general, spin canting…The existence of an inversion center between adjacent spin centers can result in the disappearance of antisymmetric exchange [33]. …in a central symmetric crystal system…m-pym group.” has been revised to “Spin canting is generally attributed…different sublattices…m-pym group.” to refer referee’s suggestion.

2.      P9, L238-242. The sentence has been divided in to two sentences as referee’s suggestion.

3.      P8, L222. The chi*T value has been changed to 1.00 cm3 mol-1 K as referee’s suggestion.

4.      P3. The numbering in the Table 2 has been revised to coincide with the numbering in Figure 1 as to refer referee’s suggestion.

5.      P2, L60. The “previously” has been changed to “previous’.

6.      P6, L193-194. The sentence “To the best of our knowledge, compound 3 cappears to be the first example of a metal coordination polymer with a bbe topology.” has been addressed to “To the best of our knowledge, compound 1 appears to be the first example of a metal coordination polymer with a bbe topology.” as refer referee’s suggestion.

7.      P8, L215. The “smooth” has been changed to “smoothly”.

8.      P12. L314. The “pyrimidy” has been changed to “pyrimidyl”.

Round 2

Reviewer 1 Report

Comment:

1) Line 222

Two Br- anions (Br1, Br2 and Br3) or are Br1 and Br2?

2) Line 229

The value of angle 40.20(40)º is correct or is 40.20(4). Revise all values

3) Line 234-235

Delete: to each otcher

4) Line 448

Delete: of the

Author Response

 P5, L146-149.The sentence “where four nitrogen atoms… two Br- anions (Br1, Br2 and Br3 and their symmetrical equivalents) …” has been revised to “where four nitrogen atoms (N6, N7 and their symmetrical equivalents for Ni1; N3, N5 and their symmetrical equivalents for Ni2) from the bpym ligands and two Br- anions (Br1, Br2 for Ni1; Br3 and its symmetrical equivalent for Ni2) occupy equatorial and axial positions, respectively.” to refer referee’s suggestion.

P5, L155; The “40.20(40)o” has been changed to “40.20(4)o” as referee’s suggestion.

P5, L161. The “to each octher” has been revised to “to each other” as referee’s suggestion.

P1, L44-45. The sentence “However, …because the ligand lacks flexibility because of the strong chelating effects.” has been revised to “However, CP’s from 2,2'-bipyrimidine usually have a low-dimensional structure (0D, 1D and 2D) because the ligand lacks flexibility by the strong chelating effects.” to refer referee’s suggestion.

Reviewer 2 Report

It is very sorry that the EPR experiment at low temperature is not succeed because of the limitation of EPR instrument accessory. However, EPR spectra with a g-value ~2.0 at 77K shows the existence of a paramagnetic spins in Ni complexes. I expect this signal will be changed to a signal with a g-value ~3 - ~4 near a transition temperature. Anyway, I will agree to publish in this journal "polymers".   

Author Response

Thanks for referee’s comment. I agree referee’s mention. As cooling to Tc, the EPR signal of a weak-ferromagnetic material, compound 1, will change from a isotropic (g ~ 2.0) to an anisotropic (g ~3 to 4) due to change of distribution of anisotropy axes.